# Advances in the Study of Hyperprogression of Different Tumors Treated with PD-1/PD-L1 Antibody and the Mechanisms of Its Occurrence

**DOI:** 10.3390/cancers15041314

**Published:** 2023-02-18

**Authors:** Jianpei Zheng, Xueyuan Zhou, Yajuan Fu, Qi Chen

**Affiliations:** Fujian Key Laboratory of Innate Immune Biology, Biomedical Research Center of South China, Fujian Normal University Qishan Campus, College Town, Fuzhou 350117, China

**Keywords:** immune checkpoint, PD-1/PD-L1, cancer, hyperprogression disease, immunotherapy

## Abstract

**Simple Summary:**

Immune checkpoint inhibitors have shown great clinical success in treating patients with various forms of cancer in recent years. However, as immune checkpoint inhibitors have become more widely used, a side effect known as hyperprogression has been recognized, which can accelerate tumor growth in patients and be life-threatening. Here, we discuss the current therapeutic status of PD-1/PD-L1 antibodies, the occurrence of HPD in various types of tumors, and the underlying mechanisms. It is hoped that researchers will investigate the mechanisms involved in HPD in greater depth to achieve better tumor treatment outcomes.

**Abstract:**

Immune checkpoint inhibitors (ICIs) including PD-1/PD-L1 antibodies, have demonstrated significant clinical benefits in the treatment of individuals with many types of cancer. However, as more and more patients use such therapies, the side effects of immune checkpoint inhibitors have also been discovered. These include accelerated tumor growth in some patients, creating new lesions, and even life-threatening ones. These side effects are known as hyperprogression disease (HPD), and different types of tumors have different HPD conditions after ICIs treatment. Therefore, understanding the pathogenesis of HPD and predicting its occurrence is critical for patients using ICIs therapy. Here, we will briefly review the current status of PD-1/PD-L1 antibody therapy, HPD occurrence in various types of tumors, and the underlying mechanism.

## 1. Introduction

Cancer is one of the most severe diseases that endanger human health. Under normal circumstances, the body’s immune system can identify and eliminate tumor cells; however, tumor cells have an “immune escape” mechanism that allows them to evade the immune system’s recognition and attack in various ways, allowing them to proliferate in the body and prevent elimination. Immune checkpoints control the strength and extent of immune responses by sending inhibitory signals to T cells, preventing damage and death of normal tissues. Immune checkpoints have become one of the primary sources of immune tolerance during tumor formation and development. Malignant tumor cells could produce immune checkpoint ligands that block effector T cell functions, thereby avoiding detection and clearance by the immune system. As receptor-ligand interaction regulates these immune checkpoints, specific monoclonal antibodies have been developed to inhibit them and treat cancer.

Immune checkpoint inhibitors (ICIs) are often monoclonal antibodies that destroy malignant tumor cells by inhibiting co-suppressive signaling pathways to reactivate the immune response and increase cytotoxic CD8+T lymphocytes [1]. In clinical treatment, ICI therapies have been used for many patients, from back-line to first-line and from single to multi-drug combinations; many patients with advanced-stage cancer have shown clinical benefits from these drugs [2]. The study found that with an increase in patients with cancer receiving ICI therapies; the condition of a fraction of patients worsens and survival is shortened by immunotherapy, leading to cancer cell proliferation. This pattern of tumor reaction, known as hyperprogression disease (HPD), is a potentially dangerous adverse effect of immune checkpoint blockade treatment. The most effective and widely used immunomodulatory drugs in clinical practice are against the immune checkpoint PD-1. Various cancers express a PD-1 ligand, PD-L1, and exploit the PD-L1/PD-1 signaling pathway to avoid T-cell immunity and promote immunological escape [3]. Here, we review the HPD phenomenon and mechanisms that have emerged in recent years due to the use of PD-1/PD-L1 antibodies for the treatment of various kinds of cancer.

## 2. The Current State of PD-1/PDL1 Treatment

### 2.1. PD-1/PD-L1 Signaling Pathway

PD-1, an immunosuppressive transmembrane protein belonging to the CD28 superfamily, is expressed by T lymphocytes. PD-1 has two ligands, PD-L1 and PD-L2 [4,5]. PD-L1 is expressed in many types of cells, primarily hematopoietic and non-hematopoietic cells, and is induced by pro-inflammatory cytokines. SHP-2, a protein tyrosine phosphatase, mediates the inhibitory properties of PD-1. PD-L1 binding to PD-1 triggers the recruitment of SHP-2 to the T cell receptor (TCR), inducing the phosphorylation of the downstream signaling molecules, such as spleen tyrosine kinase (SYK) and phospholipid inositol-3-kinase (PI3K), leading to inhibition of T cell growth, differentiation, and cytokine secretion [6]. To protect tissues from autoimmune disorders, T-cell responses need to be suppressed through a negative feedback mechanism whereby pro-inflammatory factors stimulate the upregulation of PD-1 ligand expression to protect the organism from immune attack by cytotoxic T lymphocytes (CTL), a phenomenon known as “adaptive immune resistance” [7]. Based on animal studies, PD-1 pathway blockade could have different effects on different tumors. Studies using PD-1^−/−^ mice showed that some tumors are resistant to PD-1 blockade or clearance, while others are susceptible to loss of PD-1 signaling [8,9].

### 2.2. Clinical Utility of PD-1/PD-L1 Monoclonal Antibodies

A variety of advanced cancers are currently treated with ICIs, such as melanoma (MEL), small cell lung cancer (SCLC), non-small cell lung cancer (NSCLC), renal cell carcinoma (RCC), advanced gastric cancer (AGC), head and neck squamous cell carcinoma (HNSCC), classical Hodgkin’s lymphoma (CHL), colorectal cancer (CRC), hepatocellular carcinoma (HCC), Merkel cell carcinoma (MCC) and urothelial carcinoma (UC). These drugs are different from typical cytotoxic therapies and molecularly targetable drugs, of which most ICIs kill tumor cells by targeting immunological synapses and revitalizing T cells. ICIs show significant survival benefits in the treatment of multiple cancer types; particularly, PD-1 inhibitors have long-lasting benefits due to the immune system’s memory, providing a period of five to ten years without tumor recurrence or progression and long-term patient survival [10]. As a result, ICIs therapy has become one of the most successful cancer treatments in humans [11].

### 2.3. Current Status of PD-1/PD-L1 Antibodies

Please see Table 1.

## 3. Hyperprogression Disease (HPD) and Pseudoprogression (PP)

### 3.1. Definition of HPD

According to the response evaluation criteria in solid tumors (RECIST) 1.1 criteria, a significant increment in the dimensions of the tumor lesion, or the appearance of new lesions, is considered a definite indication of disease progression. A small percentage of patients with tumors inevitably develop atypical symptoms following ICIs treatment, including HPD and pseudoprogression (PP), and are most common in patients with advanced tumors treated with ICIs. Champiat et al. were among the first to report this phenomenon, showing that 9% (*n* = 131) of patients who received anti-PD-1/PD-L1 antibody therapy exhibited HPD [12].

The assessment of HPD is primarily based on tumor growth rate (TGR) or tumor growth kinetics (TGK), defined as the percent increase in tumor volume over a month. TGR is estimated in immunotherapy by adding the maximum diameter of tumor lesions and the period between examinations, offering a dynamic, quantitative assessment of the tumor, and assisting doctors in confirming if the given treatment suppresses tumor development. HPD is defined as a rapid increase in the rate of tumor development caused by ICIs. Current criteria for HPD include therapeutic failure in less than 2 months, increased tumor load greater than 50% [13], accelerated tumor growth kinetics, and a more than threefold increase in tumor progression within two months of immunotherapy initiation compared to pre-treatment imaging [14].

### 3.2. Definition of PP

PP is an immunity-related condition in some patients during their treatment with ICIs. Some of its symptoms include an increase in tumor lesions or the development of novel lesions followed by a decrease in tumor burden due to tumor necrosis or inflammatory cell infiltration. PP is characterized by a decrease and then a transient increase in tumor load due to an inflammatory response of the immune system to immune checkpoint therapy, resulting in an increase in tumor size that is evident by imaging [15]. This increase in the tumor size due to inflammation can be misunderstood as HPD. The tumor responds effectively to treatment once the inflammation has subsided. PP might appear due to an inflammatory reaction generated by tumor immune cell invasion or postponed immune response. Currently, only a few cases of PP have been reported.

The concept of PP was initially used to describe brain tumors treated with temozolomide medication. Temozolomide is a pro-drug that acts through DNA methylation. Brain tumors have been shown to grow in size until they respond to temozolomide treatment [16]. In a research study on the anti-CTLA-4 antibody ipilimumab for MEL, ICI treatment was found to be associated with PP for the first time [17]. Further, PP was observed with the PD-1 blockers pembrolizumab and nivolumab [18]. In clinical trials, the incidence of PP in NSCLC, HNSCC, UC, mesothelioma, RCC, and MCC were 0.6–5.8, 1.8, 1.5–7.1, 6.9, 5.7–8.8, and 1.1%, respectively [19]. Given that we frequently discontinue effective treatment at the start of clinical treatment due to the presence of PP, it is vital to distinguish between PP and HPD.

### 3.3. Distinguish between HPD and PP

Contrary to PP, tumor outgrowth in HPD is produced by the particular action of ICIs as tumor promoters rather than as inflammation. HPD and PP can be distinguished by examining circulating tumor DNA (ctDNA) levels [20]. Free DNA is extracted from the patient’s plasma for liquid biopsy to test changes in ctDNA levels and uncover changes in tumor-specific replication and genomic instability. However, HPD and PP cannot be distinguished in all the patients using this method, necessitating further research into the mechanisms involved.

## 4. Effects of PD-1/PD-L1 Inhibitor Therapy in the Occurrence of HPD in Different Types of Tumors 

### 4.1. In Non-Small Cell Lung Cancer (NSCLC)

Ferrara et al. tracked the tumor progression of 406 patients with NSCLC after treatment by PD-1/PD-L1 antibodies from 2011 to 2017 [21]. They found that HPD occurred in 13.8% of these individuals and that most of these patients developed metastatic tumor lesions and had low clinical survival rates. In recent years, Lahmar et al. also conducted a retrospective examination of patients with NSCLC and discovered that 10% had acquired HPD [22].

In HPD patients treated with ICIs, cytotoxic CD8+ T lymphocytes in the tumor microenvironment are depleted due to dysregulation, and hence the elevated expression of depletion signals such as TIGIT, CTLA-4, and TIM-3, leads to lower expression of cytokines such as IFN-γ and TNF-α, weakening anti-tumor immunity [23,24]. Kim et al. collected peripheral blood samples from 144 (18.9%—45/237) patients with NSCLC before therapy and discovered low numbers of effector memory CD8+ T cells in the peripheral blood of patients with HPD and increased severe depletion CD8+ T cells with high expression of PD-1 and TIGIT [25]. These findings imply that CD8+T cell depletion is one of the probable mechanisms through which ICI treatment increases tumor growth, and may predict HPD based on the severity of T cell depletion.

In another study, Russo et al. evaluated 152 individuals with NSCLC and found that 39 (25.7%) developed HPD after receiving ICIs [26]. After processing tissue samples from all patients with HPD, researchers discovered many CD33, PD-L1, and CD163-positive tumor-associated macrophages (TAM). TAM can activate and remodel these cells by interacting with Fc receptors and PD-1 inhibitors, which initiate immunosuppressive signals that induce HPD in a subset of patients.

Gainor et al. discovered that low response rates to PD-1/PD-L1 inhibitor therapy in patients with NSCLC were associated with epidermal growth factor receptor (EGFR) mutations or anaplastic lymphoma kinase rearrangements (ALK) [27]. EGFR, a member of the HER family of human epidermal growth factor receptors, is a receptor for epithelial growth factor required for cell proliferation and signaling; mutations or overexpression of EGFR can cause tumors [28].

Mutations in EGFR that cause HPD may result in innate immunity, upregulation of cytostatic receptors, proinflammatory cytokines, immunosuppressive cells, and tumors suppressing T cell function. This is achieved by activating the PD-1/PD-L1 pathway and increasing pro-inflammatory cytokines to suppress anti-tumor immunity and promote immune escape [29].

### 4.2. In Melanoma (MEL)

Champiat et al. [12] examined 45 patients with advanced MEL and discovered that four (9%) had HPD. Schuiveling et al. [30] examined the clinical and radiological findings of all patients with advanced MEL treated with ICIs from 2013 to 2019, and showed that these patients rarely exhibit excessive tumor progression. HPD or suspected HPD was found in 142 (1.2%) patients with advanced MEL receiving ICIs therapy, including one treated with anti-PD-1 and one with anti-PD-1 in combination with anti-CTLA-4. 

Barham et al. [31] reported a patient with advanced MEL who developed both HPD and fulminant myocarditis following dual therapy with anti-PD-1 and anti-CTLA-4. Therefore, increased anti-PD-1 and anti-CTLA-4-induced inflammatory T-cell activation may lead to tumor resistance mechanisms and carditis. They also found that tumor-infiltrating CD8+ T cells lacked tumor-killing capacity. 

### 4.3. In Advanced Sarcoma (AS)

Klemen et al. [32] followed 134 individuals with advanced sarcoma (AS) over an extended period from 2015 to 2019. Of these, 69 patients (51.5%) had stable disease or achieved partial or even complete remission, 45 patients (33.6%) had disease progression but no HPD, and 15 (11.2%) had HPD. These findings imply that PD-1 inhibition mediates long-term responses in patients with advanced-stage sarcomas.

### 4.4. In Renal Cell Carcinoma (RCC) and Urothelial Carcinoma (UC)

Hwang et al. [33] investigated 203 patients with urogenital cancer treated with PD-1/PD-L1 antibodies between 2015 and 2018, of whom 102 had RCC and 101 had UC. They showed that HPD occurred in 11.9% of patients with UC, all having creatinine >1.2 mg/dL. In contrast, only 0.9% of patients with RCC had HPD, demonstrating that HPD occurred primarily in patients with UC, while the incidence of HPD in patients with RCC was minimal.

### 4.5. In Advanced Gastric Cancer (AGC)

Sasaki et al. found that 13 (21%) of 62 patients with AGC treated with ICIs experienced HPD [34]. In another study, 112 patients with AGC were recruited and treated with an anti-PD-1 antibody (nivolumab or pembrolizumab), and the remaining 57 were treated with irinotecan monotherapy. Irinotecan interacts with DNA topoisomerase I, which relieves torsional strain in DNA by inducing reversible single-strand breaks. About 10% (12/112) of patients experienced HPD following anti-PD-1 antibody therapy [35]. Age, gender, neutrophil-to-lymphocyte ratio (NLR), and lactate dehydrogenase (LDH) were uncovered as clinical factors that differed between AGC patients with and without HPD. LDH levels were not substantially correlated with the development of HPD. In contrast, low baseline albumin levels of 3.25 mg/dL were found to be significantly associated with the development of HPD. By binding to inflammatory chemicals, albumin can reduce the concentration of intensely active free cytokines. Systemic inflammation can lead to decreased synthesis and even breakdown of albumin, which can be enhanced by IL-6 treatment [36,37]. Hypoalbuminemia is widespread in patients with cancer, and has been proposed as a prognostic biomarker for immunotherapeutic treatment.

Therefore, HPD can produce various clinical characteristics and biomarkers in AGC and other malignancies, presumably because of differences in inflammation in each tumor type.

### 4.6. In Hepatocellular Carcinoma (HCC)

One study enrolled 189 patients with advanced HCC who were receiving nivolumab treatment [38]. A four-fold rise in the TGK to TGR ratio and a 40% increase in TGR were used to define HPD; 12.7% (24/189) of patients treated with nivolumab satisfied these criteria, indicating that a small proportion of patients with advanced HCC develop HPD after treatment with anti-PD-1 antibodies. Another study investigated 69 patients who received anti-PD-1 therapy between 2017 and 2020and discovered that HPD occurred in 10 cases (14.49%) [39].

### 4.7. In Head and Neck Squamous Cell Carcinoma (HNSCC)

Saada-Bouzid et al. identified 34 patients with HNSCC treated with anti-PD-L1/PD-1 drugs, 10 (29%) of whom acquired HPD [14]. Another study examined 46 individuals with HNSCC who received anti-PD-1/PD-L1 antibody monotherapy or anti-PD-L1 and anti-CTLA-4 combination therapy [40]. Of these, 18 (39%) individuals satisfied the HPD criteria. It was reported that PD-1 is expressed in NSCLC tumor cells and that PD-1 inhibition may accelerate cancer progression [41]. 

### 4.8. In Colorectal Cancer (CRC)

Chen et al. discovered that five (22.7%) of 22 patients with CRC developed HPD after anti-PD-1 antibody therapy [42]. They further showed that patients with KRAS mutations were more likely to develop HPD, presumably because KRAS-driven lung cancer frequently inactivates the STK11 gene, resulting in a poor response to immunotherapy [43]. Recent studies show that concurrent STK11 and KRAS mutations can be used as potential biomarkers to predict HPD; however, the results are limited by the small number of patients investigated [44].

### 4.9. In Lymphoma

There is a scarcity of information on the prevalence of HPD in lymphoid malignancies. A study showed that HPD could develop in four out of twelve patients with peripheral T-cell lymphoma (PTCL) treated with nivolumab [45]. Furthermore, three of the patients with HPD had angioimmunoblastic T-cell lymphoma (AITL), a T-cell lymphoma arising from follicular helper T-cells (TFH) in which the tumor cells frequently express PD-1 [46]. Blocking with anti-PD-1 antibodies may accelerate tumor growth and hence HPD.

### 4.10. In Breast Cancer (BC)

Depending on the patient and the tumor, the anti-tumor capacity may differ between PD-1 and PD-L1 inhibitors. Feng et al. reported that a 67-year-old woman with BC rapidly progressed to HPD after two cycles of chemotherapy combined with the anti-PD-1 antibody pembrolizumab. She subsequently received the PD-L1 inhibitor atezolizumab in combination with chemotherapy to alleviate her clinical symptoms [47]. Dendritic cells (DCs), where PD-L1 binds two receptors, PD-1 and CD80, are a crucial target of anti-PD-L1 antibodies (B7.1). By reducing PD-L1 expression on DCs, PD-L1 inhibitors ease PD-L1 segregation from CD80 in cis structures, allowing CD80/CD28 contact to enhance T-cell activation [48]. PD-L1 blockade allows DCs to reassert their function to generate effective T-cell anti-tumor immunity.

BC susceptibility genes, BRCA, consist of two genes, BRCA1 and BRCA2. BRCA1 and BRCA2 are oncogenes whose protein products are engaged in processes such as DNA repair [49]. BRCA2 mutational enrichment is linked to HPD [50]. The lack of function of BRCA2′s six key mutant protein structure domains may disrupt the repair of double-stranded DNA breaks and homologous recombination. This is associated with an ICIs response and may lead to HPD.

Please see Table 2 for the incidence of HPD in various types of cancer.

## 5. Pathological Factors of HPD

As described above, the prevalence of HPD ranges from 0.9% to 39%. Different cancers respond differently to ICIs treatment, and the biomarkers of HPD in them may also vary. As a result, biomarkers of HPD for various cancer types must be identified to avoid the deleterious effects of ICIs, which have substantial clinical implications for increasing patient survival and improving quality of life. In the following, we summarize the recent progress regarding the prevalence, clinical relevance, and molecular causes of HPD produced by ICIs.

### 5.1. Biomarkers of HPD

Champiat et al. and Saada-Bouzid et al. found no link between baseline tumor load, prior treatment, tumor histology, immunotherapy type, or the number of metastatic locations with HPD [12,14]. Several clinical variables, such as age and gender, were thought to be associated with HPD. Specific genetic alterations are also linked to HPD, such as MDM2/4 amplification and EGFR gene abnormalities.

Using biomarkers to provide a basis for the occurrence or imminent occurrence of HPD during the course of treatment, and to target patients, is crucial for avoiding HPD caused by ICIs treatment. Therefore, the search for biomarkers of HPD is essential for understanding and combating HPD now and in the future.

Genomic mutations could act as biomarkers for immunotherapy and HPD. Kato et al. used sequencing to examine the genomic profiles of 155 patients with cancer. They found that HPD was associated with MDM2/4 and EGFR mutations, particularly MDM2/4 amplification, which occurred in 67% (4/6) of patients, and that two patients with MDM2/4 mutation amplification had tumors that deteriorated rapidly and developed new metastases [13].

The tumor suppressor gene p53 is usually inactivated in human malignancies, whereas the oncogene MDM2 is overexpressed. MDM2 causes p53 to degrade and lose its tumor suppressive function [51], most likely by activating the JAK-STAT signaling pathway as a result of ICIs treatment, promoting the expression of IRF-8, which binds to the MDM2 promoter and causing MDM2 overexpression [52,53]. In patients treated with ICIs, further amplification of their MDM2 gene results in loss of function of the p53 tumor suppressor activity, leading to apoptosis and cell cycle arrest, which in turn accelerates tumor progression and the occurrence of HPD. Therefore, MDM2/4 amplification might be a useful diagnostic marker for HPD. 

For other relevant biomarkers of HPD, please see Figure 1.

### 5.2. Possible Causes of HPD

#### 5.2.1. HPD and Clinical Variables

Champiat et al. reported that 9% of patients with cancer they recruited could be identified as having HPD, and HPD is considered age-related, with older patients having worse overall survival compared to younger patients [12]. This could be due to changes in T-cell quantity, variety, phenotype, and function, as T-cell immunity declines with age [54]. Additionally, T-cell signaling via the TCR decreases with age [55,56].

In terms of gender, Kanjanapan et al. examined 182 patients with cancer, of whom 146 received PD-1/PD-L1 antibodies as monotherapy and 36 received a combination of immunotherapy and other therapies. Among them, 12 (6.6%) had HPD, with 2.0% (2/99) being male and 12.0% (10/83) being female [57].

#### 5.2.2. Activation of the P38 Pathway in Macrophages

Many years ago, Prehn [58] introduced the idea of tumor immunostimulation, in which the anti-cancer immune response is bidirectional, with robust immune responses suppressing tumors, weak immune responses accelerating tumor growth, and extremely weak responses having no impact. This suggests that ICIs therapy may result in inadequate immune responses in the tumor immunosuppressive microenvironment, encouraging tumor growth and the development of HPD. Studies have shown that a poor anti-tumor immune response in macrophages recruited at tumor sites with higher activation of the p38 signaling pathway could induce tumor growth [59]. Further studies have shown M2 macrophage infiltration in the tumor microenvironment of patients with HPD [26,60]. Montagna et al. found that treatment with ICIs produced a weak anti-tumor immune response that may contribute to the development of HPD. They further showed that combining ICIs with specific inhibitors of mesotyrosine and p38 resulted in significant tumor suppression [61].

#### 5.2.3. PD-1+ Treg Amplification

Studies have shown that PD-1+ Treg can cause the induction of HPD. Regulatory T cells (Treg) are a subclass of CD4+ cells that specifically express Forkhead box protein 3 (Foxp3) and are critical in regulating the anti-tumor response of T cells in the tumor microenvironment. Suppression of PD-1 expression could lead to increased expression of other immune checkpoints (such as CTLA-4, LAG-3, and TIGIT) or activation and proliferation of Tregs, leading to impaired immunomodulating capacity [62]. Foxp3 is involved in the induction of immune tolerance to Treg. Sasidharan et al. found that the monoclonal antibody against pembrolizumab could hinder Treg proliferation and lower Foxp3 expression in vitro via the mTOR pathway [63]. Weber et al. found that Treg cells were elevated in non-responders and decreased in responders in patients with malignant MEL treated with PD-1 monoclonal antibodies [64]. Later, Kamada et al. found that mice with Treg cells deficient in PD-1 could inhibit effector T cell anti-tumor responses and advanced tumor growth, while PD-1 blockade promoted the proliferation of highly suppressive PD-1+ effector Treg cells in the tumor microenvironment, leading to HPD [65]. In vivo experiments by Zhang et al. revealed that PD-1-deficient Treg cells were more immunosuppressive and provided stronger defence against autoimmune diseases than PD-1-integrated Treg cells [66], suggesting that PD-1 is involved in the increased immunosuppressive activity of Treg cells. Therefore, PD-1 immunosuppression may stimulate the proliferation of PD-1+ Treg cells in the tumor microenvironment via the TCR signaling pathway, enhancing Treg cells’ immunosuppressive role. HPD might occur when Treg cells are more prevalent than CD8+T cells in tumor tissues, favouring rapid tumor progression [67]. Furthermore, growing PD-1+ Treg cells can take up IL-2 and remove IL-2 from tumor-reactive effector T cells, leading to insufficient IL-2 required for effector T cell activation, further reducing effector T cell functions [68]. PD-L1 can also inhibit autocrine T-cell immunity by boosting Treg cell formation and maintaining Treg cell activity.

#### 5.2.4. Tumor-Infiltrating T Cells

Immune checkpoints are inhibitory signals that acts as “brakes” to prevent T cells from being overly excited when immune cells are activated. Immune checkpoint blockade therapy based on PD-1/PD-L1 restores T-cell function by preventing the binding of PD-1 or PD-L1 antibodies during ICI treatment, and promotes anti-tumor immunity. Patients who do not respond to ICIs treatment exhibit CD8+ T-cell infiltration in their tumor microenvironment. However, a recent study discovered that IFN- produced by T cells in HPD patients stimulates enhanced tumor FGF2 signaling, which ultimately reprograms tumor stem cells and leads to HPD via a sequence of signaling events [69].

It has been demonstrated that PD-L1 is expressed in both CD4+ and CD8+ T cells in the tumor microenvironment and that PD-L1+ T cells in the tumor microenvironment inhibit nearby PD-1+ T cells and promote macrophage tumor tolerance, allowing for excessive tumor progression [70]. One article reported that in a patient with lung adenocarcinoma, who developed HPD after treatment with a PD-L1 blocking antibody, duvalumab, the vast majority of infiltrating CD8+ T cells in the tumor microenvironment were expressing PD-L1 in large amounts [71]. The presence of PD-L1+CD8+ T cells thereby inhibits the anti-tumor effects of adjacent PD-1+CD8+ T cells. This may be a reason for the excessive tumor progression observed in anti-PD-L1 therapy. 

#### 5.2.5. Surrogate Rise in Other Immune Checkpoints

HPD might also be triggered by T-cell depletion caused by compensatory mechanisms of other immune checkpoints. T cells can express various immune checkpoints other than PD-1, such as CTLA-4, LAG-3, and TIM-3. T cells are often exposed to prolonged antigenic or inflammatory signals in chronic infections and malignancies, causing them to become “depleted”. Depleted T lymphocytes lose their potent effector function and express various inhibitory receptors. A study analysed the immune microenvironment of tumors treated with PD-1 inhibitors in a lung adenocarcinoma mouse model and found an upregulation in alternative immune checkpoints of T cells bound to PD-1 antibodies, particularly TIM-3, in mice with tumor progression after PD-1 antibody treatment. The expression of TIM-3 continued to increase with ICIs treatment. At the same time, no elevation of TIM-3 was detected in positive control cases [72]. PD-1 inhibitor-mediated upregulation of TIM-3 could deplete tumor-infiltrating lymphocytes (TILs) and generate tumor escape by phosphatidylinositol 3 kinase (PI3K)/Akt complexes downstream of the TCR signaling. TILs could upregulate TIM-3 in a PI3K/Akt-dependent manner and lead to tumor escape upon PD-1 blockade [73]. When PD-1 immunosuppressive treatment fails, TIM-3 antibodies may facilitate survival. Similarly, PD-1 inhibition boosted CTLA-4 and LAG-3 expression on cytotoxic CD8+T cells in an ovarian cancer model [62]. These results show that when PD-1 is blocked, other immune checkpoints can upregulate and compensate for it, resulting in, for example, suppression of T-cell activation and depletion, which eventually leads to HPD.

## 6. Conclusions

While immunotherapy has improved the overall survival of a minority of patients with metastatic/advanced disease in various cancer types, clinical symptoms of PP and HPD have created complications for the therapy. It is still unclear how HPD is caused by PD-1/PD-L1 antibodies, and the subject remains highly controversial. HPD is known to occur in almost all malignancies following ICIs treatments, and is associated with poor prognosis. As the underlying mechanisms and regulators are unknown, which limits the clinical use of ICIs, it is critical to understand the pathogenesis and preventative measures of HPD.

In summary, as shown in Figure 2, HPD is associated with MDM2/4 gene amplification, which activates the JAK-STAT signaling pathway by elevating IFN-γ after treatment with ICIs. This could lead to the induction of IRF-8 binding to the promoter of MDM2, further amplifying MDM2, leading to P53 protein degradation, and ultimately triggering HPD. Furthermore, compensatory upregulation of other immune checkpoints could lead to CD8+ T cell depletion and PD-L1+ T cells suppress neighbouring PD-1+ T cells, resulting in a diminished anti-tumour effect, expansion of Foxp3+ PD-1+ Treg cells after ICI treatment and uptake of IL-2 to suppress effector T-cell function, BRCA2 gene mutations, TAM aggregation, activation of the p38 pathway, and EGFR gene mutations to increase pro-inflammatory cytokines such as IL-6, IL-10, and TGF-β.

Although the above biomarkers have been implicated inHPD, none are absolute, and different tumor types have varied immune responses to ICIs. Hence, the identification of more biomarkers is required for more reliable determination and prediction of HPD. In-depth research is required on the immune checkpoint signaling pathways and the mechanisms associated with HPD. Examining specific effects on the tumor microenvironment and the interaction among or between immune cells and tumor cells following immune checkpoint treatments, identifying new biomarkers associated with HPD, and the interactions between different immune checkpoints would help identify and predict HPD, resulting in better treatment outcomes.

## Figures and Tables

**Figure 1 cancers-15-01314-f001:**
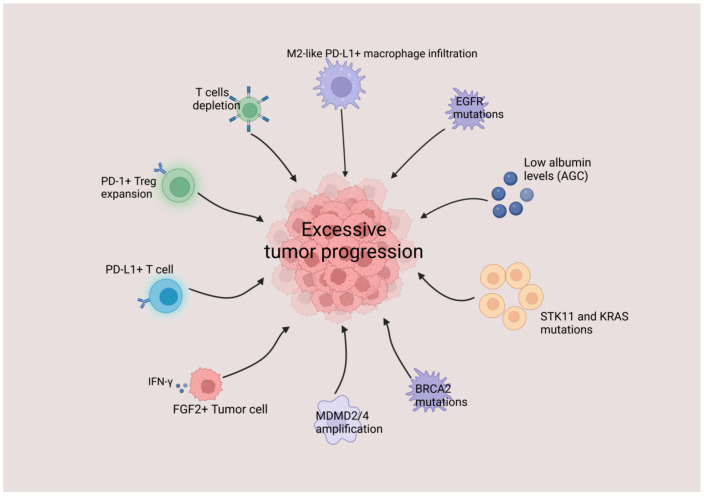
Possible biomarkers of HPD.

**Figure 2 cancers-15-01314-f002:**
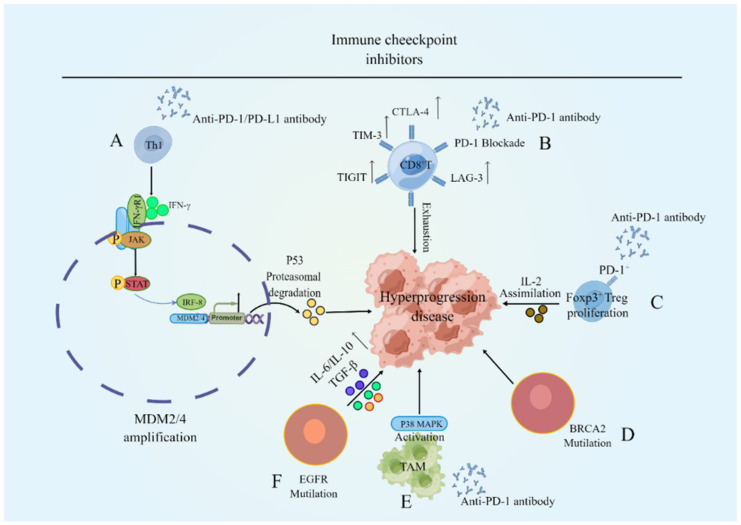
The mechanisms of HPD. (**A**) ICIs treatment causes Th cells to oversecrete IFN-γ, causing IRF-8 expression in the JAK-STAT pathway to be activated. IRF-8 binds to the MDM2/4 promoter, causing the MDM2/4 gene to be amplified, resulting in the loss of p53 activity and the acceleration of tumor growth. (**B**) When PD-1 is blockaded, T cells in the tumor microenvironment upregulate the expression of other immunological checkpoints such as CTLA-4, LAG-3, and TIM-3, resulting in tumor development. (**C**) Immunosuppression of Foxp3+PD-1+ Treg cells is lifted following PD-1 blocking, allowing Treg cells to proliferate and take up IL-2 while inhibiting effector T cell anti-tumor actions. (**D**) ICIs treatment in breast cancer may result in BRCA2 oncogene mutations and the development of HPD. (**E**) ICIs produce weak anti-tumor responses, resulting in increased activation of the p38 signaling pathway and tumor growth stimulation. (**F**) Mutations in EGFR drive innate immune upregulation of T cell suppressor receptors PD-1/PD-L1 and CTLA-4 expression, causing T cells to secrete pro-inflammatory cytokines such as IL-6, IL-10, and TGF-β and immune escape to occur.

**Table 1 cancers-15-01314-t001:** PD-1/PD-L1 antibody.

Name	R&D Company	Application	Cycle/Dose
Keytruda (pembrolizumab)	Merck	MEL, NSCLC, NHSCC, CHL, UC, CRC, AGC, MSI-H/DMMR, Cervical Cancer	Once every three weeks/2 mg/kg
Opdivo (Nivolumab)	Bristol-Myers Squibb	MEL, NSCLC, SCLC, NHSCC, CHL, UC, CRC, RCC, HCC	Once every fortnight/3 mg/kg
Lambrolizumab	Merck	MEL, NSCLC, RCC	Once every fortnight/10 mg/kg
Treprizumab	TopAlliance	MEL, UC, Nasopharyngeal Carcinoma	Once every fortnight/3 mg/kg
Carrelizumab	Hearem	NSCLC, HCC, Esophageal Cancer, Nasopharyngeal Carcinoma, and Lymphoma.	Once every fortnight/200 mg
sintilimab	Innovent	CHL, NSCLC, HCC	Once every three weeks/200 mg
Tislelizumab	BeiGene	CHL, UC, NSCLC, HCC	Once every three weeks/200 mg
Tecentri (Atezolizumab)	Roche	UC, NSCLC	Once every three weeks/1200 mg
Bavencio (avelumab)	Pfizer and Merck	MCC, Bladder Cancer	Once every fortnight/10 mg/kg
Imfinzi (Durvalumab)	AstraZeneca	UC, NSCLC	Once every fortnight/10 mg/kg
Tilelizumab	BeiGene	CHL, UC, NSCLC, HCC, Solid tumors, Nasopharyngeal Carcinoma, Esophageal squamous carcinoma	Once every fortnight/200 mg
Cemiplimab	Regeneron Pharmaceuticals	Metastatic cutaneous squamous cell carcinoma, Basal cell carcinoma, NSCLC	Once every three weeks/350 mg

**Table 2 cancers-15-01314-t002:** The prevalence of HPD in various cancer types.

Cancer Types	Reference	Incidence of HPD
Non-small-cell lung cancer (NSCLC)	Ferrara et al. [21]	13.8% (56/405)
	Lahmar et al. [22]	10% (9/89)
	Kim et al. [25]	18.9% (45/237)
	Russo et al. [26]	25.7% (39/152)
Melanoma (MEL)	Champiat et al. [12]	9% (4/45)
	Schuiveling et al. [30]	1.2% (2/142)
Advanced sarcoma (AS)	Klemen et al. [32]	11.2% (15/134)
Renal cell carcinoma (RCC)	Hwang et al. [33]	0.9% (1/102)
Urothelial carcinoma (UC)	Hwang et al. [33]	11.9% (12/101)
Advanced gastric cancer (AGC)	Sasaki et al. [34]	21% (13/62)
	Kim et al. [35]	10.7% (12/112)
Hepatocellular carcinoma (HCC)	Kim et al. [38]	12.7% (24/189)
	Zhang et al. [39]	14.49% (10/69)
Head and neck squamous cell carcinoma (NHSCC)	Saada-Bouzid et al. [14]	29% (10/34)
	Economopoulou et al. [40]	39% (18/46)
Colorectal cancer (CRC)	Chen et al. [42]	22.7% (5/22)
Lymphoma	Bennanl et al. [45]	33.3% (4/12)

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
