# Peer review of "Advances in the Study of Hyperprogression of Different Tumors Treated with PD-1/PD-L1 Antibody and the Mechanisms of Its Occurrence"

_cancers, 2023, doi:10.3390/cancers15041314_

Round 1

Reviewer 1 Report

This review briefly outlines the current status of PD-1/PD-L1 antibody therapy, HPD occurrence in various types of tumors, and the underlying mechanism. From Pubmed, I didn’t search the similar review. The review generally well written, and had the novelty, but the language was not authentic, and should be revised. The following questions need to answer:

1.       When you cited the other’s papers, you should use your languages to describe the published results or conclusions, such as lines 133, 136,141, etc.

2.       Figure.1 should add the legend, and I suggest add another new figure to summarize the biomarkers of HPD.

3. Many wrong typos and grammar should be revised.

4. Some references were old, and some format was missed, such as Ref.23.

Reviewer 2 Report

The authors report on the current status of checkpoint inhibitor therapy with PD-1/PD-L1 antibodies and the occurrence of hyperprogression disease, which displays are serve side effect of checkpoint inhibitor therapy. They very well summarized the prevalence of hyperprogression disease in different tumors like non-small-cell lung cancer, melanoma but also gastric, renal liver and colon cancer. As a suggestion, the authors may consider also status of T cell infiltration and activation during checkpoint inhibitor therapy, since T cells are the main effector cells.

Some minor phrases could be revised e.g.

 Line 12:

… growth acceleration …

 Line 32

… used …     should be deleted

 Line 50

.. is produced by T lymphocytes. … Better “expressed by”
